# Gender differences influence over insomnia in Korean population: A cross-sectional study

Yun Kyung La[1], Yun Ho Choi[2], Min Kyung Chu[3], Jung Mo Nam[4], Young-Chul Choi[1], Won-Joo Kim[1] *

1 Department of Neurology, Gangnam Severance Hospital, Yonsei University College of Medicine, Seoul, Republic of Korea, 2 Department of Neurology, Incheon St. Mary's Hospital, College of Medicine, The Catholic University of Korea, Incheon, Republic of Korea, 3 Department of Neurology, Severance Hospital, Yonsei University College of Medicine, Seoul, Republic of Korea, 4 Department of Preventive Medicine, Severance Hospital, Yonsei University College of Medicine, Seoul, Republic of Korea

* kzoo@yuhs.ac

## Abstract

### Study objectives

Insomnia is the most common sleep disorder with significant psychiatric/physical comorbidities in the general population. The aim of this study is to investigate whether socioeconomic and demographic factors are associated with gender differences in insomnia and subtypes in Korean population.

### Method

The present study used data from the nationwide, cross-sectional study on sleep among all Koreans aged 19 to 69 years. The Insomnia Severity Index (ISI) was used to classify insomnia symptoms and their subtypes (cutoff value: 9.5). The Pittsburgh Sleep Quality Index (PSQI), Goldberg Anxiety Scale (GAS) and Patient Health Questionnaire-9 (PHQ-9) were used to measure sleep quality, anxiety and depression.

### Results

A total of 2695 participants completed the survey. The overall prevalence of insomnia symptoms was 10.7%, including difficulty in initiating sleep (DIS) (6.8%), difficulty in maintaining sleep (DMS) (6.5%) and early morning awakening (EMA) (6.5%), and these symptoms were more prevalent in women than in men. Multivariate analysis showed that female gender, shorter sleep time and psychiatric complications were found to be independent predictors for insomnia symptoms and subtypes. After adjusting for covariates among these factors, female gender remained a significant risk factor for insomnia symptoms and their subtypes. As for men, low income was related to insomnia.

### Conclusion

Approximately one-tenth of the sample from the Korean general population had insomnia symptoms. The prevalence of insomnia symptom and the subtypes were more prevalent in women than men. Gender is an independent factor for insomnia symptoms.

**Data Availability Statement:** All relevant data are within the manuscript and its Supporting Information files.

**Funding:** This study was supported by a 2011 grant from Korean Academy of Medical Sciences

and Korean Neurological Association (grant # KNA-10-MI-03). Funders had no role in the study design, data collection and analysis, decision to publish, or preparation of the manuscript.

**Competing interests:** Yun Kyung La has no conflicts of interest to declare. Choi Yun Ho has no conflicts of interest to declare. Jung Mo Nam has no conflicts of interest to declare. Young-Chul Choi has no conflicts of interest to declare. Won-Joo Kim has no conflicts of interest to declare. Min Kyung Chu was involved as a site investigator for a multicenter trial sponsored by Otsuka Korea, Novartis International AG and Eli Lilly and company. Min Kyung Chu worked an advisory member for Teva and received lecture honoraria from Allergan Korea and Yuyu Pharmaceutical Company in the past 24 months. This does not alter our adherence to PLOS ONE policies on sharing data and materials.

**Abbreviations:** CBTI, Cognitive Behavioral Therapy for Insomnia; DIS, difficulty in initiating sleep; DMS, difficulty in maintaining sleep; EMA, early morning awakening; GAS, Goldberg Anxiety Scale; IS, insomnia symptoms; ISI, The insomnia severity index; PHQ-9, Patient Health Questionnaire-9; PSQI, Pittsburgh Sleep Quality Index.

# Introduction

Insomnia is the most common sleep disorder in the general population. It is associated with impaired social performance, cultural difference and daytime functioning, along with other psychological/physical conditions [1–3]. Insomnia affects 6–18% of the general population [4–7]. Therefore, insomnia imposes a significant personal and social burden [8]. Insomnia prevalence varies according to the definition of insomnia or insomnia symptom [9].

Different insomnia symptoms have been defined as subtypes of insomnia. These subtypes include difficulty in initiating sleep (DIS), difficulty in maintaining sleep (DMS) and early morning awakening (EMA) [9]. There are differences in the prevalence, association with excessive daytime sleepiness and psychiatric comorbidities among these subtypes [10].

Gender, age, socioeconomic status and psychiatric comorbidities are known to be significant factors for insomnia prevalence [9]. The elderly population has an increased insomnia prevalence compared to the young or middle age population in almost all epidemiological studies [11–13]. Insomnia prevalence is stable from 15 to 44 years of age and increases over 45 years [9]. Insomnia prevalence is typically higher among individuals with lower incomes and lower education levels [14]. However, some studies have reported contradictory results [15, 16]. The association between insomnia and psychiatric disorders has been repeatedly demonstrated. Approximately 90% and 80% of individuals with anxiety and depression had insomnia symptoms in cross-sectional studies, respectively [17, 18]. Longitudinal studies confirmed the close association of anxiety and depression with insomnia [16, 19].

Short sleep time and poor sleep quality are significant factors for insomnia. A survey in six European countries showed that a significant proportion of individuals with insomnia had short sleep times either voluntarily or non-voluntarily [20]. The authors classified sleep deprivation as a subtype of insomnia. Poor sleep quality is also common among individuals with insomnia [21].

Female gender has been recognized as a significant factor for insomnia. Epidemiological studies have consistently shown a higher prevalence of insomnia symptoms among women compared to men [22–24]. The difference in insomnia prevalence between women and men increases with age [9]. In addition to insomnia or insomnia symptoms, women report more frequent dissatisfaction with sleep and the daytime consequences of insomnia [4, 22, 24].

Nevertheless, information regarding gender differences in insomnia subtypes is currently limited. In addition, gender differences in insomnia symptoms adjusting for significant covariates have rarely been reported. The purpose of this study is to investigate gender differences in insomnia symptoms and their subtypes regarding covariates using population data in South Korea.

# Methods

## Study population and survey process

The present study population was collected from cross-sectional surveys. In brief, it was a nationwide, cross-sectional study on sleep among Koreans aged 19 to 69 years. We used a two-stage clustered random sampling method proportional to the population and socioeconomic distribution for all Korean territories. The study protocol was approved by the institutional review board/ethics committee of Hallym University Sacred Heart Hospital in Korea(Approval No. 2011-I077), conducted according to the principles expressed in the Declaration of Helsinki.; Participants provided their written informed consent to participate in this study. Trained interviewers conducted structured interviews using a questionnaire to assess sleep time, sleepiness, anxiety and depression using a face-to-face interview. The interview included

questions on sleep status. To minimize potential interest bias, we informed candidates that the topic of the survey was social health issues rather than sleep issues. All interviewers were employed by Gallup Korea and had previous social survey interviewing experience.

## Assessment for sleep time

We asked all participants to report their usual sleep time in terms of hours and minutes, separately for workdays and free days, during the past month. The average sleep time was a weighted mean of the sleep time on workdays and free days, calculated as ([workday sleep time x 5] + [free day sleep time x 2))/7. Short sleep time was defined as an average sleep time of <6 h [25].

## Assessment for insomnia symptom and poor sleep quality

The Insomnia Severity Index (ISI) was used to classify insomnia symptoms. The ISI is a self-reporting, brief screening measure of insomnia that contains questions corresponding in part to the diagnostic criteria of insomnia [26]. The ISI comprises seven items concerning the severity of sleep-onset difficulty, sleep-maintaining difficulty, early awakening and satisfaction with sleep patterns. Each item was rated on a 0–4 scale [27]. The ISI was previously validated with good sensitivity and specificity and showed a 9.5 cut-off score for discriminating patients with "insomnia symptoms" in the population-based sample [28]. Among those who satisfy this definition (ISI score $\geq$ 10), we further classified the subtypes of insomnia symptoms as difficulty in initiating sleep (DIS), difficulty in maintaining sleep (DMS) and early morning awakening (EMA) if a participant responded with $\geq$ 2 on the scale (intermediate or higher) for those items [20]. The Pittsburgh Sleep Quality Index (PSQI) was used to measure sleep quality. If a participant's PSQI score was 6 or higher, she/he was classified as having poor sleep quality [29].

## Anxiety and depression assessment

We used the Goldberg Anxiety Scale (GAS) to diagnose anxiety among participants. The GAS questionnaire was composed of nine items: four screening items and five supplementary items. Anxiety was diagnosed when there were positive answers to two or more screening items and five or more of all scale items [30]. The Korean version of the scale has a sensitivity of 82.0% and specificity of 94.4% for diagnosing anxiety [31].

The Patient Health Questionnaire-9 (PHQ-9) was used to diagnose depression [32]. It is composed of nine items each scored 0 to 3. Participants who scored 10 or higher on this measure were considered to have depression. The Korean PHQ-9 has a sensitivity of 81.8% and specificity of 89.9% [33].

## Statistical analysis

The Kolmogorov–Smirnov test was used to confirm the normality of the distribution. The Student's t-test was used to compare continuous variables after the normality of the sample was confirmed. Post hoc analyses were performed using Tukey's method. Categorical variables were compared using the Chi-square test. The significance level was set at $p < 0.05$ for all analyses.

We calculated the odds ratios (ORs) with 95% confidence intervals (CIs) for the occurrence of insomnia symptoms and their subtypes through univariable and multivariable logistic regression analyses. In univariable analyses, we modelled the ORs for insomnia symptoms without adjusting for covariates. In multivariable analyses, we used four models. In Model 1, adjustment was conducted for sociodemographic variables (age, gender, size of residential area

and educational level) and short sleep time. Model 2 incorporated anxiety (GAS score) with Model 1. Model 3 included depression (PHQ-9 score ≥ 10) with Model 1. The final model, Model 4, incorporated poor sleep quality (PSQI score ≥ 6), anxiety and depression with Model 1. Statistical analyses were performed using the Statistical Package for Social Sciences 23.0 (IBM, Armonk, NY, USA).

## Results

### Survey

The interviewers approached 7,430 individuals and 3114 of them agreed to take the survey. After 352 individuals suspended the interview, 2695 participants completed the survey (cooperation rate of 36.2%; Fig 1). The distributions of age, gender, size of residential area and educational level were not significantly different from those of the general population of Korea. (Table 1).

### Prevalence of insomnia and poor sleep quality

Of the 2695 participants, 290 (10.7%) were classified as having insomnia symptoms according to the ISI score. Seven-hundred and fifteen (26.5%) participants who reported PSQI scores higher than 5 were classified as having poor sleep quality. 182 (6.8% of all participants) were accompanied by DIS, 175 (6.5%) with DMS and 176 (6.5%) with EMA (Fig 2).

### Demographic characteristics, psychiatric comorbidities and short sleep time according to the presence of insomnia symptom

The mean age of participants with insomnia symptoms did not significantly differ from participants without insomnia symptoms (43.8 ± 14.2 vs. 42.9 ± 13.7, p = 0.823). The proportion of

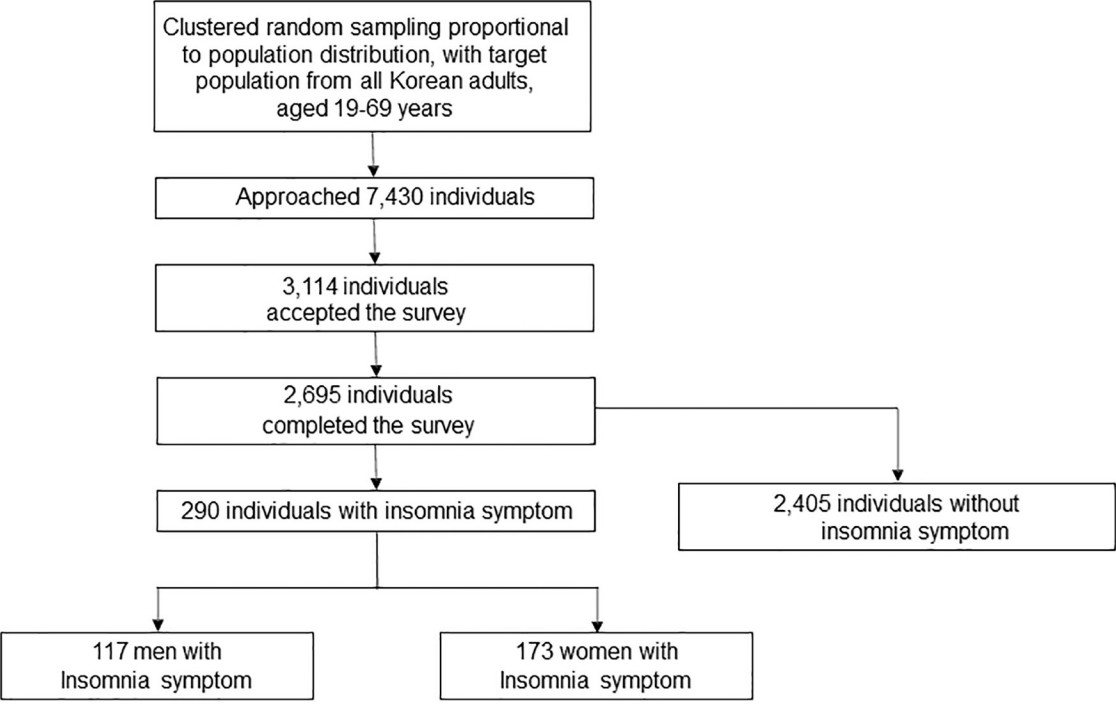

**Fig 1. Flow chart depicting the participation of subjects in the Korean headache-sleep study.**

**Table 1. Sociodemographic characteristics of survey participants; the total Korean population; and cases identified as having insomnia.**

| | Survey participants N (%) | Total population N (%) | P | Poor sleep quality N, % (95% CI) | p | Insomnia N, % (95% CI) | p | Short sleep time | p | Average sleep time (hours + SD) | P |
|---|---|---|---|---|---|---|---|---|---|---|---|
| **Sex** | | | | | | | | | | | |
| Men | 1,345 (49.3) | 17,584,365 (50.6) | 0.854[a] | 334, 24.8 (22.5–27.1) | 0.046 | 117, 8.7 (7.2–10.2) | 0.001 | 141, 10.5 (8.8–12.1) | 0.727 | 7.3 + 1.2 | 0.109 |
| Women | 1,350 (50.7) | 17,198,350 (49.4) | | 381, 28.2 (25.8–30.6) | | 173, 12.8 (11.0–14.6) | | 136, 10.1 (8.5–11.7) | | 7.3 + 1.2 | |
| **Age** | | | | | | | | | | | |
| 19–29 | 542 (20.5) | 7,717,947 (22.2) | 0.917[a] | 153, 28.3 (24.4–32.0) | 0.028 | 59, 10.9 (9.2–12.6) | 0.426 | 45, 8.3 (6.0–10.6) | 0.002 | 7.6 + 1.3 | <0.001 |
| 30–39 | 604 (21.9) | 8,349,487 (24.0) | | 136, 22.5 (19.2–25.9) | | 53, 8.8 (7.3–10.3) | | 42, 7.0 (4.9–9.0) | | 7.5 + 1.1 | |
| 40–49 | 611 (23.1) | 8,613,110 (24.8) | | 167, 27.3 (23.8–30.1) | | 66, 10.8 (9.1–12.5) | | 73, 11.9 (9.4–14.5) | | 7.1 + 1.2 | |
| 50–59 | 529 (18.9) | 6,167,505 (17.7) | | 160, 30.2 (26.3–34.2) | | 63, 11.9 (10.2–13.6) | | 70, 13.2 (10.3–16.1) | | 7.1 + 1.3 | |
| 60–69 | 409 (15.6) | 3,934,666 (11.3) | | 99, 24.2 (20.0–28.4) | | 49, 12.0 (10.2–13.7) | | 47, 11.5 (8.4–14.6) | | 7.1 + 1.2 | |
| **Size of residential area** | | | | | | | | | | | |
| Large city | 1,248 (46.3) | 16,776,771 (48.2) | 0.921[a] | 338, 27.1 (24.6–30.0) | 0.541 | 136, 10.9 (9.2–12.6) | 0.945 | 131, 10.5 (8.8–12.2) | 0.670 | 7.3 + 1.2 | 0.471 |
| Medium-to-small city | 1186 (44.0) | 15,164,345 (43.6) | | 303, 25.5 (23.1–28.0) | | 125, 10.5 (8.9–12.2) | | 116, 9.8 (8.1–11.5) | | 7.3 + 1.2 | |
| Rural area | 261 (9.7) | 2,841,599 (8.2) | | 74, 28.4 (22.8–33.9) | | 29, 11.1 (9.4–12.8) | | 30, 11.5 (7.6–15.4) | | 7.3 + 1.3 | |
| **Education** | | | | | | | | | | | |
| Middle school or Less | 393 (14.9) | 6,608,716 (19.0) | 0.752[a] | 110, 28.0 (23.5–32.4) | 0.917 | 62, 15.8 (13.8–17.7) | 0.006 | 47, 12.0 (8.7–15.2) | 0.514 | 7.2 + 1.4 | 0.012 |
| High school | 1,208 (44.5) | 15,234,829 (43.8) | | 317, 26.2 (24.0–28.7) | | 116, 9.6 (8.0–11.2) | | 127, 10.5 (8.8–12.3) | | 7.2 + 1.3 | |
| College or more | 1,068 (39.6) | 12,939,170 (37.2) | | 281, 26.3 (23.7–29.0) | | 109, 10.2 (8.6–11.8) | | 100, 9.4 (7.6–11.1) | | 7.4 + 1.3 | |
| Not responded | 26 (9.6) | | | 7, 26.9 (8.7–45.2) | | 3, 1.0 (0.9–1.3) | | 3, 11.5 (0.0–24.7) | | 7.4 +1.2 | |
| **Total** | 2695 (100.0) | 34,782,715 (100.0) | | 715, 26.5 (24.8–28.2) | | 290, 10.7 (9.1–12.4) | | 277, 10.3 (9.1–11.4) | | 7.4 + 1.3 | |

[a]Comparison of gender, age group, size of residential area, and educational level distributions between the sample in the present study and the total population of Korea

*Abbreviations*: CI = confidence interval

women was higher among participants with insomnia symptoms than those without insomnia symptoms (59.6% vs. 49.1%, p < 0.001). The prevalence of anxiety and depression was significantly higher among participants with insomnia symptoms than in those without (40.3% vs. 6.3%, p < 0.001 for anxiety and 25.9% vs. 1.7%, *p* < 0.001 for depression). The average sleep time of all participants was 7.4 ± 1.3 h.

Two-hundred and seventy-seven (10.3%) participants were classified as having short sleep time. The average sleep time of participants with insomnia symptoms was significantly shorter than that of participants without insomnia symptoms (6.7 ± 1.5 h vs. 7.4 ± 1.2 h, *p* < 0.001). The prevalence of short sleep time (< 6 hours of average sleep time) was higher among participants with insomnia symptoms (35.5% vs. 15.2%, *p* < 0.001; Table 2).

## Prevalence of insomnia symptoms in women and men

A higher percentage of women reported insomnia symptoms compared to men (12.8% vs. 8.7%, p = 0.001). The mean ISI score was significantly higher in women compared to men

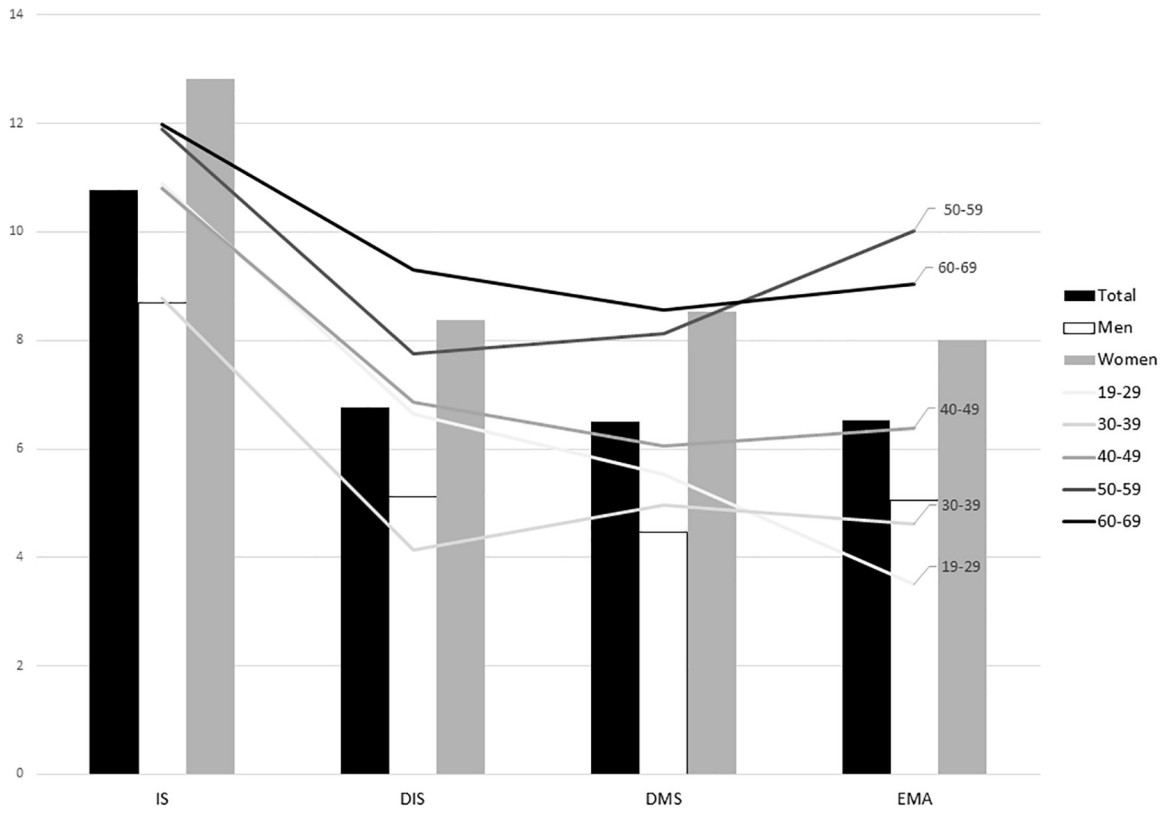

**Fig 2. Prevalence of insomnia symptoms in men and women with different age groups.** *p* was calculated by χ2 test. *Abbreviations*: IS = Insomnia Symptoms, DIS = Difficulty in Initiating Sleep; DMS = Difficulty in Maintaining Sleep; EMA = Early Morning Awakening.

(4.2 ± 4.9 vs. 3.3 ± 4.2, *p* < 0.001). All subtypes of insomnia symptoms including DIS (8.3% vs. 5.1%, *p* < 0.001), DMS (8.5% vs. 4.4%, *p* < 0.001) and EMA (8.0% vs. 5.0%, *p* = 0.001) were more prevalent in women than men (Fig 2).

## Insomnia severity of women and men among participants with insomnia

Among 290 participants with insomnia symptoms, the total ISI score was not significantly different between women and men (14.6 ± 4.4 vs. 14.0 ± 4.3, *p* = 0.199). The prevalence of DIS (37.6% vs. 33.3%, *p* = 0.460), DMS (40.5% vs. 32.5%, *p* = 0.168) and EMA (36.4% vs. 29.9%, *p* = 0.251) did not significantly differ between women and men.

## Factors associated with insomnia symptoms in women and men

When comparing participants with insomnia symptoms to participants without insomnia symptoms, age, size of residential area and education level did not significantly affect the prevalence of insomnia. In contrast, female gender (OR = 1.3, 95% CI = 1.0–1.78), short sleep time (OR = 2.6, 95% CI = 1.9–3.5), anxiety (OR = 6.0, 95% CI = 4.3–8.3) and depression (OR = 9.6, 95% CI = 6.0–15.3) were found to be independent predictors of insomnia symptoms in multiple regression analysis (Table 3).

For DIS, female gender (OR = 1.4, 95% CI = 1.0–2.0), short sleep time (OR = 3.1, 95% CI = 2.1–4.4), anxiety (OR = 5.0, 95% CI = 3.4–7.3) and depression (OR = 9.6, 95% CI = 6.0–15.5) were found to be independent predictors (S1 Table).

**Table 2. Distribution of demographic, social and lifestyle factors according to the presence of insomnia symptoms.**

|  | Subjects with IS N = 290 | Subjects without IS N = 2405 | P-value |
|---|---|---|---|
| **Demographics** |  |  |  |
| Mean age ± SD (years) | 43.76±14.20 | 42.88±13.70 | 0.366 |
| Women, N (%) | 173 (59.6) | 1182 (49.1) | 0.001 |
| **Education level** |  |  |  |
| Middle school or less | 62 | 331 | 0.006 |
| High school | 116 | 1092 |  |
| College or more | 109 | 959 |  |
| Not responded | 3 | 23 |  |
| **Residential area** |  |  |  |
| Large city | 136 | 1112 | 0.943 |
| Medium-to-small city | 125 | 1061 |  |
| Rural area | 29 | 232 |  |
| **Smoking, N (%)** | 78 (26.9) | 666 (27.7) | 0.486 |
| **Alcohol, N (%)** | 185 (63.8) | 1597 (66.4) | 0.313 |
| **ISI score, ± SD** | 14.37±4.39 | 2.47±2.50 |  |
| **Anxiety, N (%)** | 117 (40.3) | 151 (6.3) | <0.001 |
| **Depression, N (%)** | 75 (25.9) | 41 (1.7) | <0.001 |
| **BMI, ± SD (kg/cm^2)** | 23.09± 3.30 | 22.94±2.97 | 0.031 |
| **Average sleep duration, ± SD, N (short sleep duration)** | 6.68± 1.54 (103) | 7.36± 1.17 (366) | <0.001 |
| **Poor sleep quality, N (%)** | 221 (76) | 494 (20.5) | <0.001 |

*p* was calculated by χ2 test or student's t test.

*Abbreviations*: IS–Insomnia symptoms

**Table 3. Univariable and multivariable regression analysis for insomnia symptoms.**

|  | Univariable ORs | | Multivariable analysis ORs | | | | | | | |
|---|---|---|---|---|---|---|---|---|---|---|
|  |  |  | Model 1 | | Model 2 | | Model 3 | | Model4 | |
|  | OR (95%Ci) | p-value | OR (95%Ci) | p-value | OR (95%Ci) | p-value | OR (95%Ci) | p-value | OR (95%Ci) | p-value |
| **Sex (Women)** | 1.543 (1.204–1.977) | 0.001 | 1.513 (1.172–1.953) | 0.001 | 1.401 (1.068–1.839) | 0.015 | 1.388 (1.057–1.824) | 0.018 | 1.327 (1.000–1.760) | 0.05 |
| **Age (40 years or older)** | 1.199 (0.934–1.539) | 0.155 | 0.928 (0.702–1.226) | 0.599 | 0.938 (0.698–1.261) | 0.673 | 1.028 (0.762–1.387) | 0.856 | 0.999 (0.734–1.361) | 0.997 |
| **Size of residential area (Large city)** | 1.027 (0.804–1.311) | 0.831 | 1.037 (0.807–1.332) | 0.779 | 1.004 (0.768–1.312) | 0.977 | 1.010 (0.771–1.321) | 0.944 | 0.990 (0.749–1.308) | 0.943 |
| **Education (Middle school or less)** | 1.707 (1.260–2.314) | 0.001 | 1.552 (1.105–2.180) | 0.011 | 1.427 (0.993–2.051) | 0.055 | 1.556 (1.083–2.235) | 0.017 | 1.473 (1.009–2.150) | 0.045 |
| **Sleep duration (6 hours or shorter)** | 3.069 (2.354–4.000) | <0.001 | 3.081 (2.348–4.042) | <0.001 | 2.738 (2.039–3.675) | <0.001 | 2.780 (2.069–3.736) | <0.001 | 2.604 (1.913–3.544) | <0.001 |
| **Anxiety** | 10.095 (7.579–13.447) | <0.001 |  |  | 9.284 (6.910–12.473) | <0.001 |  |  | 6.014 (4.344–8.325) | <0.001 |
| **Depression** | 20.113 (13.409–30.170) | <0.001 |  |  |  |  | 18.626 (12.243–28.336) | <0.001 | 9.588 (6.026–15.257) | <0.001 |

In Model 1, adjustment was conducted for sociodemographic variables (age, sex, size of residential area and educational level) and short sleep time. Model 2 incorporated anxiety (GAS score) with Model 1. Model 3 included depression (PHQ-9 score ≥ 10) with Model 1. The final model, Model 4, incorporated poor sleep quality (PSQI score ≥ 6), anxiety and depression with Model 1. Subject with missing data was excluded from the analysis.

*p* was calculated by the univariable / multiple logistic regression analysis.

*Abbreviations*: OR = odds ratio, CI = confidence interval.

**Table 4. Univariable and multivariable regression analysis for insomnia symptoms and insomnia subtypes in men.**

| | Multivariable analysis ORs | | | | | | | |
| --- | --- | --- | --- | --- | --- | --- | --- | --- |
| | IS | | DIS | | DMS | | EMA | |
| | OR (95%Ci) | p-value | OR (95%Ci) | p-value | OR (95%Ci) | p-value | OR (95%Ci) | p-value |
| **Age (40 years or older)** | 0.772 (0.479–1.244) | 0.287 | 1.204 (0.653–2.222) | 0.552 | 1.220 (0.630–2.362) | 0.555 | 1.469 (0.767–2.816) | 0.246 |
| **Size of residential area (Large city)** | 1.433 (0.919–2.235) | 0.113 | 1.420 (0.809–2.493) | 0.222 | 1.104 (0.607–2.010) | 0.745 | 0.932 (0.523–1.659) | 0.810 |
| **Education (Middle school or less)** | 0.869 (0.411–1.836) | 0.713 | 0.598 (0.236–1.517) | 0.279 | 0.865 (0.362–2.071) | 0.745 | 0.810 (0.345–1.900) | 0.628 |
| **Sleep duration (6 hours or shorter)** | 3.489 (2.177–5.591) | <0.001 | 3.664 (2.055–6.533) | <0.001 | 3.247 (1.756–6.006) | <0.001 | 4.855 (2.724–8.654) | <0.001 |
| **Anxiety** | 6.031 (3.517–10.341) | <0.001 | 6.059 (3.215–11.419) | <0.001 | 9.307 (4.113–21.061) | <0.001 | 6.849 (3.630–12.922) | <0.001 |
| **Depression** | 18.158 (8.267–39.882) | <0.001 | 12.766 (5.677–28.711) | <0.001 | 5.578 (2.855–10.900) | <0.001 | 7.614 (3.306–17.536) | <0.001 |
| **Monthly income (<,2000 dollar)** | 2.030 (1.130–3.645) | 0.018 | 2.006 (0.988–4.074) | 0.054 | 2.776 (1.363–5.653) | 0.005 | 2.477 (1.231–4.984) | 0.011 |
| **Occupation (unemployed)** | 1.174 (0.644–2.141) | 0.600 | 1.487 (0.708–3.123) | 0.295 | 0.798 (0.338–1.881) | 0.606 | 0.834 (0.360–1.928) | 0.671 |

Subject with missing data was excluded from the analysis.

p was calculated by the univariable / multiple logistic regression analysis.

*Abbreviations*: OR = odds ratio, CI = confidence interval, IS = Insomnia Symptoms, DIS = Difficulty in Initiating Sleep, DMS = Difficulty in Maintaining Sleep, EMA = Early Morning Awakening

In univariable analysis, age was a significant predictor of DIS (1.5, 95% CI = 1.1–2.1) and DMS (OR = 1.5, 95% CI = 1.1–2.0). However, age lost significance for DIS (p = 0.235) and DMS (*p* = 0.219) in multiple regression analysis. Female gender (OR = 1.8, 95% CI = 1.2–2.6), short sleep time (OR = 2.5, 95% CI = 1.7–3.7), anxiety (OR = 5.2, 95% CI = 3.6–7.7) and depression (OR = 8.6, 95% CI = 5.3–13.9) were significant predictors of DMS in multiple regression analysis (S2 Table).

In multiple regression analysis for EMA, female gender (OR = 1.4, 95% CI = 1.0–2.0), age (OR = 1.8, 95% CI = 1.2–2.6), educational level (OR = 1.6, 95% CI = 1.1–2.5), short sleep time (OR = 3.7, 95% CI = 2.6–5.3), anxiety (OR = 5.1, 95% CI = 3.4–7.6) and depression (OR = 5.7, 95% CI = 3.4–9.4) were independent predictors (S3 Table).

Women showed similar result aspects. In multiple regression analysis for insomnia symptoms, short sleep time (OR = 2.2, 95% CI = 1.4–3.3), anxiety (OR = 6.1, 95% CI = 4.0–9.3), and depression (OR = 6.9, 95% CI = 3.8–12.3) were significant predictors. Those factors also had significance for insomnia symptoms, DIS, DMS and EMA in multiple regression analysis for both genders. Additionally, relatively low monthly income was a significant risk factor for men in insomnia symptoms and their subtypes (Table 4), while no socio-economic risk factor was related to female insomnia. (S4 Table).

## Discussion

The main findings of the present study were as follows: 1) The prevalence of insomnia symptoms in a Korean general population-based sample was 10.7% and 2) The prevalence of insomnia symptoms was significantly higher in women than men. Among the subtypes of insomnia symptoms, there was a higher prevalence of DIS, DMA and EMA in women than in men and 3) Female gender was a significant factor for insomnia symptoms even after adjusting for covariates including sociodemographic variables, short sleep time, anxiety and depression.

General insomnia prevalence was similar with that reported in previous study [34]. Gender differences in insomnia disorder or insomnia symptoms have also been demonstrated in epidemiological studies. A meta-analysis combining data from 29 studies reported that women were at a 41% greater risk for having insomnia than men in adult populations [22]. Our results are in agreement with the previous findings that women had more insomnia symptoms than men.

What are possible mechanisms for higher insomnia prevalence in women? One possible explanation is roles of sex steroids. The major gonadal sex steroids are estrogen and progesterone in women and testosterone in men [35]. Sleep complaints typically co-occur in women with the fluctuation of ovarian steroids such as puberty, pregnancy, the menstrual cycle and the menopausal period [36–38]. Estrogen replacement therapy improves sleep disturbances in menopausal women [37]. In men, androgen deprivation therapy worsens insomnia in prostate cancer patients [38, 39]. On the contrary, high-dose testosterone replacement was associated with a reduction of sleep efficacy and total sleep time [40]. In summary, low or fluctuating levels of estrogen was consistently associated with increase occurrence of sleep disturbances including insomnia in women. Nevertheless, role of testosterone level on sleep in men is currently uncertain owing to inconsistent results.

Another possible explanation is role of biological gender. Androgen and estrogen affect differently on sleep between women and men. Gonadectomized female and male rats showed no significant difference in NREM and REM sleep amount. However, estradiol replacement significantly reduces NREM and REM sleep time in dark phase in female rats. In contrast, NREM and REM sleep architecture did not change by estradiol treatment in male rats [41]. Therefore, gender difference may be owing to biological gender rather than sex steroids.

The higher prevalence of mood symptoms in women could be a reason for gender difference. Both anxiety and depression are closely associated with sleep disturbances, including insomnia. Epidemiological studies have consistently reported a close association of insomnia with anxiety and depression. A meta-analysis including 21 longitudinal epidemiological studies showed an odds ratio for insomnia to predict depression of 2.60 (confidence interval: 1.98–3.42) [42]. Women preponderance of anxiety and depression has been persistently noted [43, 44]. Therefore, a higher prevalence of anxiety and depression in women may account for the higher prevalence of insomnia in women than men. Nevertheless, insomnia prevalence was significantly higher in women than men even after adjusting for anxiety and depression in the present study (Table 3). Therefore, the higher prevalence of insomnia in women could not be solely explained by the higher prevalence of anxiety and depression in women.

Differences in pain and somatic perception can also explain gender differences in insomnia symptoms. Pain and somatic symptoms are generally more prevalent in women [22, 45]. In particular, pain, insomnia, and somatic symptoms showed significant differences in adults rather than in adolescents before puberty. [46] These differences might speculate the relationship between sex and insomnia symptoms.

The present study showed that all subtypes of insomnia symptoms including DIS, DMS and EMA were more common in women than men. The significant differences in prevalence between women and men persisted even after adjusting for covariates including sociodemographic variables, short sleep time, anxiety and depression. Gender differences in the subtypes of insomnia have been demonstrated in epidemiological studies. Janson et al. reported that female gender was positively related to DMS in a European study [47]. An interview-based study showed a clear correlation between female gender and insomnia symptoms as well as the DMS-IV inclusion criteria [23]. Another study showed that 11.9% (14.0% of women vs. 9.3% of men) of the Hong Kong Chinese adult population complained of insomnia in the preceding month. Insomnia symptoms including DIS, DMS and EMA, more prominently appeared in women than in men. Gender (as women) was a specific risk factor despite common risk factors for insomnia in both genders [24].

The prevalence of EMA only showed significance in older individuals after including anxiety and depression in the adjustment (S3 Table). Still, we found a tendency that prevalence of insomnia symptoms increased with age. In this study, inclusion criteria for participants maximally limited to 69 years, and the size of the sample was small. These are possible explanations

for the absence of statistical significance. (Table 1). Gender differences in insomnia prevalence were apparent in the elderly in epidemiological studies. In elderly populations, the risk of insomnia in women was almost doubled compared to that of men. Elderly (>65 years) women had a 73% greater risk of insomnia compared to elderly men [22]. In contrast, the predisposition for insomnia in women did not appear before puberty. These findings suggest a possible role of hormonal changes in the differences in insomnia between women and men [48, 49]. Also, we failed to report lower education levels as independent risk factor for insomnia and all subtypes. The statistical significance of low monthly income in men alone offered weak explanatory power, due to disputes in recent studies [9].

Despite of the low response rate (36.3%), we can speculate the general status of the Korean population through our data: 1) there were no significant differences in the distribution of age, gender, size of residential area, and educational level between survey participants and the Korean general population. 2) The prevalence of insomnia symptoms, anxiety, depression, and short sleep duration in this sample were similar to that of previous studies [4].

The present study has some limitations. First, this study did not use any objective measurement of sleep or sleep disorders such as polysomnography or actigraphy. Nevertheless, we evaluated insomnia symptoms, sleep habits and sleep quality using validated questionnaires via face-to-face interviews. Second, we did not evaluate the use of antidepressants, anxiolytics, hypnotics or caffeine. Antidepressants, anxiolytics or hypnotics may have a positive effect on sleep time and caffeine may negatively affect sleep time. Also, we could not measure sleep-relevant-treatments such as sleep-education, relaxation training or cognitive behavioral therapy for insomnia (CBTI). Third, although the present study was a large population-based study, some subgroup analyses might not have statistical significance due to the limited sample size and low response rate. In other words, the lack of significant findings in subgroup analyses could be due to the small subgroup sample size.

The present study has several strengths. First, the present study included a large sample size across the Korean general population. The distribution of age, gender, size of residential area and educational level of our sample was similar to the Korean population. Second, the present study assessed potential covariates of insomnia symptoms such as sociodemographic variables, poor sleep quality, short sleep time, anxiety and depression, which are related to insomnia symptoms. Third, we investigated gender differences in the subtypes of insomnia symptoms including DIS, DMS and EMA in addition to overall insomnia symptoms. Further, we analyzed the factors affecting insomnia subtypes.

## Conclusions

Approximately one-tenth of the Korean general population sample had insomnia symptoms. The prevalence of insomnia symptoms was higher in women than men. Among the subtypes of insomnia symptoms, DIS, DMS and EMA were more prevalent in women than in men. The predisposition of insomnia symptoms in women was prominent even after adjusting for covariates including anxiety and depression.

## Supporting information

**S1 Table. Univariable and multivariable regression analysis for difficulty initiating sleep (DIS).** In Model 1, adjustment was conducted for sociodemographic variables (age, sex, size of residential area and educational level) and short sleep time. Model 2 incorporated anxiety (GAS score) with Model 1. Model 3 included depression (PHQ-9 score $\geq$ 10) with Model 1. The final model, Model 4, incorporated poor sleep quality (PSQI score $\geq$ 6), anxiety and depression with Model 1. Subject with missing data was excluded from the analysis. *p* was

calculated by the univariable / multiple logistic regression analysis. *Abbreviations*: OR = odds ratio, CI = confidence interval.
(DOCX)

**S2 Table. Univariable and multivariable regression analysis for difficulty maintaining sleep (DMS).** In Model 1, adjustment was conducted for sociodemographic variables (age, sex, size of residential area and educational level) and short sleep time. Model 2 incorporated anxiety (GAS score) with Model 1. Model 3 included depression (PHQ-9 score $\geq$ 10) with Model 1. The final model, Model 4, incorporated poor sleep quality (PSQI score $\geq$ 6), anxiety and depression with Model 1. Subject with missing data was excluded from the analysis. *p* was calculated by the univariable / multiple logistic regression analysis. *Abbreviations*: OR = odds ratio, CI = confidence interval.
(DOCX)

**S3 Table. Univariable and multivariable regression analysis for early morning awakening (EMA).** In Model 1, adjustment was conducted for sociodemographic variables (age, sex, size of residential area and educational level) and short sleep time. Model 2 incorporated anxiety (GAS score) with Model 1. Model 3 included depression (PHQ-9 score $\geq$ 10) with Model 1. The final model, Model 4, incorporated poor sleep quality (PSQI score $\geq$ 6), anxiety and depression with Model 1. Subject with missing data was excluded from the analysis. *p* was calculated by the univariable / multiple logistic regression analysis. *Abbreviations*: OR = odds ratio, CI = confidence interval.
(DOCX)

**S4 Table. Univariable and multivariable regression analysis for insomnia symptoms and insomnia subtypes in women.** Subject with missing data was excluded from the analysis. *p* was calculated by the univariable / multiple logistic regression analysis. *Abbreviations*: OR = odds ratio, CI = confidence interval, IS = Insomnia Symptoms, DIS = Difficulty in Initiating Sleep, DMS = Difficulty in Maintaining Sleep, EMA = Early Morning Awakening.
(DOCX)

## Acknowledgments

The authors would like to thank Gallup Korea for providing technical support.

## Author Contributions

**Conceptualization:** Yun Kyung La.

**Data curation:** Yun Kyung La, Min Kyung Chu.

**Formal analysis:** Yun Kyung La, Min Kyung Chu, Jung Mo Nam.

**Methodology:** Yun Ho Choi.

**Supervision:** Won-Joo Kim.

**Validation:** Jung Mo Nam.

**Visualization:** Jung Mo Nam.

**Writing – original draft:** Yun Kyung La.

**Writing – review & editing:** Yun Ho Choi, Min Kyung Chu, Young-Chul Choi, Won-Joo Kim.

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
