## [Decision Letter · Decision Letter 0]

30 Sep 2019

PONE-D-19-17898

Gender differences influence over insomnia in Korean population: A cross-sectional study

PLOS ONE

Dear Dr. Kim,

Thank you for submitting your manuscript to PLOS ONE. After careful consideration, we feel that it has merit but does not fully meet PLOS ONE’s publication criteria as it currently stands. Therefore, we invite you to submit a revised version of the manuscript that addresses the points raised during the review process.

Two reviewers well assessed the manuscript. 

Carefully read and respond the Reviewers' comments appropriately.

We would appreciate receiving your revised manuscript by Nov 14 2019 11:59PM. To enhance the reproducibility of your results, we recommend that if applicable you deposit your laboratory protocols in protocols.io, where a protocol can be assigned its own identifier (DOI) such that it can be cited independently in the future. For instructions see: http://journals.plos.org/plosone/s/submission-guidelines#loc-laboratory-protocols

We look forward to receiving your revised manuscript.

Kind regards,

Masaki Mogi

Academic Editor

PLOS ONE

**Journal Requirements:**

"Yun Kyung La has no conflicts of interest to declare.

Choi Yun Ho has no conflicts of interest to declare.

Min Kyung Chu was involved as a site investigator for a multicenter trial sponsored by Otsuka Korea, Novartis International AG and Eli Lilly and company. Min Kyung Chu worked an advisory member for Teva and received lecture honoraria from Allergan Korea and Yuyu Pharmaceutical Company in the past 24 months.    

Jung Mo Nam has no conflicts of interest to declare.

Young-Chul Choi has no conflicts of interest to declare.

Won-Joo Kim has no conflicts of interest to declare."

**Comments to the Author**

1. Is the manuscript technically sound, and do the data support the conclusions?

Reviewer #1: Yes

Reviewer #2: Yes

2. Has the statistical analysis been performed appropriately and rigorously? 

Reviewer #1: Yes

Reviewer #2: I Don't Know

3. Have the authors made all data underlying the findings in their manuscript fully available?

Reviewer #1: Yes

Reviewer #2: Yes

4. Is the manuscript presented in an intelligible fashion and written in standard English?

Reviewer #1: Yes

Reviewer #2: Yes

5. Review Comments to the Author

Reviewer #1: This is a cross-sectional study aiming at the gender differences of insomnia symptoms. The findings and conclusions are not new while the methodology sounds satisfactory. The findings in the current study confirmed gender differences in insomnia and its subtype in Korean population. The strengths of the study include a reasonable measurement tool and good sampling framework, despite a lack of details. I have some suggestions for the authors.

1. What is the validity for each questionnaire in Korean version? Especially for ISI and PSQI.

2. The two-stage clustered random sampling method proportional to the population and socioeconomic distribution for all Korean territories should be presented in more details. Residents from which area were randomly selected?

3. When was the study conducted?

4. The response rate was poor, which should be considered a significant limitation.

5. Gender steroids should be rephrased as “sex steroids”. In general, gender is a more sociology concept while sex is a more biology concept.

6. Another explanation for the gender difference of insomnia is the gender difference in somatic perception.

Reviewer #2: Thank you for this chance to review this interesting manuscript, titled “Gender differences influence over insomnia in Korean population: A cross-sectional study”. This manuscript reported on the prevalence and associated factors of insomnia/symptoms in a sample of 2,695 Korean Adults. It was found that insomnia symptoms were present in 10.7% of survey respondents, and were positively associated with female gender, shorter total sleep duration, and psychiatric co-morbidities/symptoms. This is an interesting study which replicates data of previous studies in the general population of Korea. Its major strengths are the large sample size and reporting of results in the Korean population. My main concern is the limited amount of novel data/findings presented in this manuscript. As this is such a potentially rich dataset and study, I’ve tried to give some general comments/questions to investigate some more novel data.

Major:

·       Non-restorative sleep is no longer included among insomnia disorder criteria (or subtype). Consider removing and focusing only on DIS, DMS, and EMA subtypes/criteria for insomnia disorder.

·       What item of the ISI was used to determine NRS? Unless an additional item was included, I don’t think any of the items assess NRS (Dis/satisfaction, Noticeable to others, worry, and daytime functioning interference).

·       Abstract: Method section, page 6 description of the ISI, etc. The ISI should not be used in isolation to ‘diagnose’ insomnia. Consider re-wording to ‘classify’ insomnia rather than ‘diagnose insomnia’ in these sections.

·       Consider selection/response bias (of the 7,430 individuals approached, is it anticipated that more/less respondents with insomnia symptoms self-selected to complete the survey?

·       Table 3. Multivariate regression analysis, examining effect of gender, while controlling for gender as co-variate? The different models/co-variates may require review from a statistical reviewer. I may have mis-read something here.

·       The last 3 items of the ISI may be confounded with measures of depression/anxiety. I wonder whether the association between the ISI and psychiatric symptoms could be driven by the overlap in symptoms assessed by these items? It may be interesting to examine a ‘night time insomnia symptom’ group, using a composite score on the first 3 or 4 ISI items only, and re-examining relationships with anxiety/depression. This could be reported in a sensitivity analysis confined to the supplement.

·       Consider a sensitivity analysis removing the “difficulties falling asleep, staying asleep, etc” question from the PHQ-9 and investigating whether the relationship between the PHQ and ISI is reduced. Shared symptoms may increase relationship here without truly reflecting a positive association between insomnia and depression.

·       Same for the ‘sleep’ items of the Goldberg Anxiety Scale.

Minor:

·       Minor grammatical changes throughout (‘symptom’ vs ‘symptoms’, etc.).

·       Consider clearly differentiating “Insomnia disorder” (chronic and frequent night-time symptoms, plus daytime functional impairments), and “Insomnia symptoms” (night time complaints only) early in the manuscript.

·       Page 9, “Prevalence of insomnia and poor sleep quality’ section – There is a potential for some wording confusion in the second and third sentence. Please consider modifying the final sentence (“182 (6.8%) were accompanied by DIS…”) to specify that the denominator was the total 2,695 participants, not the 715 participants with PSQI of 5+.

·       Consider collapsing the multiple sub-headings in the Results section into a single sub-heading (e.g. several sections report gender differences for different outcomes/definitions).

·       Supplement Tables 1, 2, correct to ‘difficulty initiating/maintaining/etc sleep’

·       Other studies have reported a greater incidence of EMA, but lower incidence of DIS in older adults. It may be interesting to make comparisons in the discussion section.

·       What is the state of treatment services for insomnia in Korea? If 10% of the population suffer from insomnia symptoms, will more resources be required to treat them? May be worth commenting on these aspects in the discussion.

Additional Considerations and other Analyses

·       Separate cohort into different age brackets and examine prevalence of different insomnia complaints (e.g. <30, 31-40, 41-50, etc.).

·       It would be interesting to present a figure of changes in the three insomnia subtypes by age.

·       Did you collect any measure of employment (either job area, work hours/week, sick leave, etc.)? This may make an interesting comparison between insomnia/no-insomnia symptoms groups.

·       Any additional measure of other medical/psychiatric co-morbidities?

·       It may be interesting to also include a different ISI cut-off to define “Moderate” or “severe” insomnia (in addition to insomnia symptoms ISI >9.5). I wonder whether your gender and age effects will persist with different definitions of insomnia.

·       Consider examining TST between weekends and weekdays in both insomnia and no-insomnia participants. Is it possible that there is a greater discrepancy between weekends/weekdays for insomnia sufferers but not for those without insomnia symptoms. It would be interesting to examine the effect of age here too.

6. PLOS authors have the option to publish the peer review history of their article (what does this mean?). If published, this will include your full peer review and any attached files.

Reviewer #1: No

Reviewer #2: No

---

## [Author Response · Author response to Decision Letter 0]

13 Nov 2019

Response to Reviewers

Reviewer #1: 

1. What is the validity for each questionnaire in Korean version? Especially for ISI and PSQI. 

Insomnia severity index : ISI≥ 10 which showed validity across population-based sample and The correlation between the ISI-K total score and PSQI-K was 0.84. A cutoff score of 15.5 on the ISI-K was optimal for discriminating patients with insomnia. (Cho YW, Song ML, Morin CM. Validation of a Korean version of the insomnia severity index. J Clin Neurol. 2014 Jul;10(3):210-5.)

Pittsburgh Sleep Quality Index : Cronbach's α coefficient for internal consistency of the total score of the PSQI-K was 0.84 which shows high reliability. Sensitivity and specificity for distinguishing poor and good sleepers were 0.943 and 0.844 using the best cutoff point of 8.5. The total and component scores of the PSQI-K for insomnia and narcolepsy were significantly higher than those for controls (p < 0.05). ( Sohn SI1, Kim DH, Lee MY, Cho YW. The reliability and validity of the Korean version of the Pittsburgh Sleep Quality Index. Sleep Breath. 2012 Sep;16(3):803-12.) which is higher than the score of 5 in the original paper (Buysse DJ, Reynolds CF 3rd, Monk TH et al. The Pittsburgh Sleep Quality Index: a new instrument for psychiatric practice and research. Psychiatry Res 28(2):193-213). This difference may be due to the high severity of symptoms in the insomnia and narcolepsy group and there is a limitation that this study was conducted on one sleep center in Korea. Consider that we conducted THE study in general population with various ages and The PSQI-K has been validated as a screening tool for sleep quality, we classified participants as “poor” sleeper if participant’s PSQI score was 6 or higher.

2. The two-stage clustered random sampling method proportional to the population and socioeconomic distribution for all Korean territories should be presented in more details. Residents from which area were randomly selected?

: Residents were randomly selected from all Korean territories except Jeju-do. 15 administrative divisions were designated as the primary sampling units. In each of the 15 administrative divisions, 4 representative basic units were randomly selected as secondary sampling units. The survey was applied in 60 representative basic units where appropriate assessments of residential status, population structure, household income, and occupational structure were available. In each sampling unit, the target sample number was determined based on the distributions of sociodemographic parameters such as age, gender, educational level, and monthly household income. And we sampled individuals in proportion to the population distribution and stratified as age, gender, and occupation. 

3. When was the study conducted?

: This survey was collected from November 2011 to January 2012. Analyzing this data, drafting and revising of this manuscript was conducted from November 2016 to July 2017.

4. The response rate was poor, which should be considered a significant limitation. 

: We agree with your comment and we described poor response rate as a limitation in the manuscript.

5. Gender steroids should be rephrased as “sex steroids”. In general, gender is a more sociology concept while sex is a more biology concept.

Referring to your opinion, we corrected the context.

6. Another explanation for the gender difference of insomnia is the gender difference in somatic perception.

We revised the manuscript based on your opinion.

Reviewer #2: 

Major:

Non-restorative sleep is no longer included among insomnia disorder criteria (or subtype). Consider removing and focusing only on DIS, DMS, and EMA subtypes/criteria for insomnia disorder.

-> Thank you for your opinion and we revised the manuscript removing NRS as subtypes for insomnia disorder.

What item of the ISI was used to determine NRS? Unless an additional item was included, I don’t think any of the items assess NRS (Dis/satisfaction, Noticeable to others, worry, and daytime functioning interference).

-> We add a question asking participants to score whether they are having none restorative sleep within the past 2 weeks from 0 (never) to 4 (severe, at least three times per week). 

Abstract: Method section, page 6 description of the ISI, etc. The ISI should not be used in isolation to ‘diagnose’ insomnia. Consider re-wording to ‘classify’ insomnia rather than ‘diagnose insomnia’ in these sections.

-> referring to your opinion, we corrected the context.

Consider selection/response bias (of the 7,430 individuals approached, is it anticipated that more/less respondents with insomnia symptoms self-selected to complete the survey?

-> To reduce sampling error, this survey was commissioned from Gallup Korea and they conducted the two-stage clustered random sampling gaining a large sample size. 

Table 3. Multivariate regression analysis, examining effect of gender, while controlling for gender as co-variate? The different models/co-variates may require review from a statistical reviewer. I may have mis-read something here.

-> Yes, as we know, it has statistical power while using gender as co-variate to examining the statistical significance of gender in multivariate regression analysis. We also asked an opinion about statistical designs of this study from Professor Jung Mo Nam, Department of Preventive Medicine.

The last 3 items of the ISI may be confounded with measures of depression/anxiety. I wonder whether the association between the ISI and psychiatric symptoms could be driven by the overlap in symptoms assessed by these items? It may be interesting to examine a ‘night time insomnia symptom’ group, using a composite score on the first 3 or 4 ISI items only, and re-examining relationships with anxiety/depression. This could be reported in a sensitivity analysis confined to the supplement.

Consider a sensitivity analysis removing the “difficulties falling asleep, staying asleep, etc” question from the PHQ-9 and investigating whether the relationship between the PHQ and ISI is reduced. Shared symptoms may increase relationship here without truly reflecting a positive association between insomnia and depression.

Same for the ‘sleep’ items of the Goldberg Anxiety Scale.

-> 

We doubt that the last three items of the ISI solely measuring anxiety or depression that can be removed. We think that ISI, PHQ-9 and Goldberg Anxiety Scale has its validity as a screening tool for each disorder/symptom as a whole questionnaire. But we agree that it might be intriguing to analysis removing “sleep section” from each questionnaire and see a correlation with insomnia symptom but since this study is focused on to see the effect of gender as a covariate in insomnia and take it into the next paper. Thank you for your opinion.

Minor:

Minor grammatical changes throughout (‘symptom’ vs ‘symptoms’, etc.).

Consider clearly differentiating “Insomnia disorder” (chronic and frequent night-time symptoms, plus daytime functional impairments), and “Insomnia symptoms” (night time complaints only) early in the manuscript.

Page 9, “Prevalence of insomnia and poor sleep quality’ section – There is a potential for some wording confusion in the second and third sentence. Please consider modifying the final sentence (“182 (6.8%) were accompanied by DIS…”) to specify that the denominator was the total 2,695 participants, not the 715 participants with PSQI of 5+.

Consider collapsing the multiple sub-headings in the Results section into a single sub-heading (e.g. several sections report gender differences for different outcomes/definitions).

Supplement Tables 1, 2, correct to ‘difficulty initiating/maintaining/etc sleep’

-> We revised the manuscript based on your opinion.

What is the state of treatment services for insomnia in Korea? If 10% of the population suffer from insomnia symptoms, will more resources be required to treat them? May be worth commenting on these aspects in the discussion.

-> We mentioned these aspects as limitations in discussion; we could not measure state of treatment services in this study. 

Additional Considerations and other Analyses

It would be interesting to present a figure of changes in the three insomnia subtypes by age.

-> We revised the manuscript based on your opinion.

Did you collect any measure of employment (either job area, work hours/week, sick leave, etc.)? This may make an interesting comparison between insomnia/no-insomnia symptoms groups. Any additional measure of other medical/psychiatric co-morbidities?

-> No, we collected only their current state of job by criteria. We gathered information on other specific medical comorbidities (hypertension, diabetes, coronary artery disease, cerebrovascular diseases, dyslipidemia, etc) and age of onset, but we ruled out these comorbidities as a covariate due to a small number of sample and low statistical power compare to others.

Separate cohort into different age brackets and examine prevalence of different insomnia complaints (e.g. <30, 31-40, 41-50, etc.).

Other studies have reported a greater incidence of EMA, but lower incidence of DIS in older adults. It may be interesting to make comparisons in the discussion section.

It may be interesting to also include a different ISI cut-off to define “Moderate” or “severe” insomnia (in addition to insomnia symptoms ISI >9.5). I wonder whether your gender and age effects will persist with different definitions of insomnia.

Consider examining TST between weekends and weekdays in both insomnia and no-insomnia participants. Is it possible that there is a greater discrepancy between weekends/weekdays for insomnia sufferers but not for those without insomnia symptoms. It would be interesting to examine the effect of age here too.

-> We did separated participants into different age groups but the results were not significantly different. The current state of employment was also irrelevant. We appreciate your great feedback and will analyze in detail and take it seriously into the next paper.

---

## [Decision Letter · Decision Letter 1]

16 Dec 2019

Gender differences influence over insomnia in Korean population: A cross-sectional study

PONE-D-19-17898R1

Dear Dr. Kim,

We are pleased to inform you that your manuscript has been judged scientifically suitable for publication and will be formally accepted for publication once it complies with all outstanding technical requirements.

With kind regards,

Masaki Mogi

Academic Editor

PLOS ONE

Additional Editor Comments (optional):

The authors well responded to the Reviewers' comments.

Reviewers' comments:

Reviewer's Responses to Questions

**Comments to the Author**

1. If the authors have adequately addressed your comments raised in a previous round of review and you feel that this manuscript is now acceptable for publication, you may indicate that here to bypass the “Comments to the Author” section, enter your conflict of interest statement in the “Confidential to Editor” section, and submit your "Accept" recommendation.

Reviewer #1: All comments have been addressed

2. Is the manuscript technically sound, and do the data support the conclusions?

Reviewer #1: Yes

3. Has the statistical analysis been performed appropriately and rigorously? 

Reviewer #1: (No Response)

4. Have the authors made all data underlying the findings in their manuscript fully available?

Reviewer #1: (No Response)

5. Is the manuscript presented in an intelligible fashion and written in standard English?

Reviewer #1: (No Response)

6. Review Comments to the Author

Reviewer #1: (No Response)

7. PLOS authors have the option to publish the peer review history of their article (what does this mean?). If published, this will include your full peer review and any attached files.

Reviewer #1: No

---

## [Editor Report · Acceptance letter]

23 Dec 2019

PONE-D-19-17898R1 

Gender differences influence over insomnia in Korean population: A cross-sectional study 

Dear Dr. Kim:

I am pleased to inform you that your manuscript has been deemed suitable for publication in PLOS ONE. Congratulations! Your manuscript is now with our production department. 

With kind regards,

on behalf of

Dr. Masaki Mogi 

Academic Editor

PLOS ONE